# Clinical Anatomy of the Ligaments of the Face and Their Fundamental Distinguishing Features

**DOI:** 10.3390/medicina60050681

**Published:** 2024-04-23

**Authors:** Artem Mirontsev, Olesya Andruschenko, Yuriy Vasil’ev, Elena Verbo, Liyana Kolesova, Ekaterina Blinova, Kirill Zhandarov, Mikhail Nelipa, Petr Panushkin, Ellina Velichko, Yulianna Enina, Zurab Bagatelia, Sergey Dydykin

**Affiliations:** 1Department of Operative Surgery and Topographic Anatomy, First Moscow State Medical University (Sechenov University), 119435 Moscow, Russiablinova_e_v@staff.sechenov.ru (E.B.); zhandarov_k_a@staff.sechenov.ru (K.Z.); panyushkin_p_v@staff.sechenov.ru (P.P.); velichko_e_v@staff.sechenov.ru (E.V.); enina_yu_i@staff.sechenov.ru (Y.E.); zurab.bagatelia@gmail.com (Z.B.); dydykin_s_s@staff.sechenov.ru (S.D.); 2Moscow Engineering Physics Institute, National Research Nuclear University MEPhI, 115409 Moscow, Russia; 3Central Research Institute of Dentistry and Maxillofacial Surgery, 119991 Moscow, Russia; olesyasurgery@gmail.com (O.A.); everbo@mail.ru (E.V.); 4Novonexus, 107076 Moscow, Russia; kolesova.liyana@gmail.com

**Keywords:** facial aging, the ligaments of the face, zygomatic ligament, zygomaticus major muscle, masseteric ligament, mandibular ligament, parotideomasseteric fascia

## Abstract

Our study aimed to clarify the anatomical features of the zygomatic, upper masseteric, lower masseteric and mandibular ligaments and their possible contribution to age-related gravitational ptosis. The study was carried out by the method of layered dissection of fresh cadavers. In several observations, the zygomatic ligament is represented by the fibers originating from the zygomaticus major muscle fibers. It is a true ligament with the fibers inserted directly into the skin. The upper and lower masseteric ligaments originate from the parotideomasseteric fascia and weave into the thickness of the SMAS. The mandibular ligament consists of two connective tissue laminae originating from the parotideomasseteric fascia at the lower edge of the mandible and from the inner surface of this fascia, along the anterior edge of the masseter muscle, skirting the facial vein sheath and the facial artery, traveling toward the platysma and the depressor anguli oris muscle, and merging with their fibers. The zygomatic ligament should be considered an osteo-musculocutaneous ligament, emphasizing the role of the associated zygomaticus major muscle in the mechanism of aging. The upper and lower masseteric and mandibular ligaments are false fascio-SMAS ligaments rather than osteo-cutaneous ones, playing the barrier role and fixing the superficial fascia and the platysma muscle.

## 1. Introduction

The concepts of the mechanisms of facial aging, which are recognized today, use age-related bone resorption as one of the key arguments, complemented by the loosening of the ligamentous apparatus of the face and the impairment of its fixing function [1]. In this regard, a significant number of bone–skin ligaments are described, which, among others, include the zygomatic, superior, and inferior masseteric, and mandibular ligaments [2]. Located on the same line, along the anterior edge of the masseter muscle, these ligaments are heterogeneous by nature. There are many discrepancies in both the descriptions and interpretations of their role [3,4,5,6]. For example, the term “McGregor’s patch” often refers to the zygomatic ligament [7,8]; however, from McGregor’s description, it appears that he meant rather the upper masseteric ligaments, usually associated with the transverse facial artery, the parotid duct, and the zygomatic branch of the facial nerve, passing through or near them. The zygomatic ligaments are described in the literature as several dense connective tissue fibers originating from the lower border of the zygomatic arch and traveling to the junction of the zygomatic arch with the body of the zygomatic bone. These ligaments are described as true ligaments, due to their attachment directly to the skin [9]. There is also a description of the zygomatic ligaments as starting and running close to the origin of the facial musculature of the area, represented by the zygomaticus major muscle [10]. Nevertheless, there is no direct statement regarding the connection between the zygomaticus major muscle and these ligaments. The masseteric ligaments are described as less robust structures extending directly from the parotideomasseteric fascia and running above the masseter muscle. There are also data on the direct origin of these ligaments from the masseter muscle [11,12]. From the description of the mandibular ligaments, it follows that they start from the anterior third of the lower jaw, piercing the posterior portion of the depressor anguli oris muscle, then they weave directly into the skin. There are significant discrepancies regarding the origin of these bundles: from two rows of parallel connective tissue fibers located 1 cm away from the lower border of the mandible to the laminae located 4.5 cm anteriorly to the angle of the lower jaw [13,14,15]. In later studies, evidence was provided that the deep part of the mandibular ligament is well distinguished having a pronounced connection with the parotideomasseteric fascia, platysma, and sites of its bone attachment, while the superficial part of the mandibular ligament is missing. 

These facts provide the basis for assumptions about the common nature of the mandibular and masseteric ligaments, allowing us to interpret all these three ligaments as fascio-SMAS ligaments. They differ from the true zygomatic ligament, the nature of which should be investigated in more detail because the literature clearly indicates the independence of this ligament from the parotideomasseteric fascia. In addition, our observations show that the zygomatic ligament is partially or completely presented by the tendon fibers of the zygomaticus major muscle, with which it has such close relations. A detailed review of the ligament from this point of view would help to complement the understanding of the dependence of the aging process on the degradation of facial muscles. 

## 2. Materials and Methods

All examinations had previously been approved by the Ethics Committee of the University of Sechenov (protocol №. 04-197 of 6 March 2019). The material of this study included the heads of 30 cadavers of adult people aged 60 to 78 years. Fifteen of them were males and fifteen were females. Since the study was performed on both halves of the head, the total amount of observations was 60. The exclusion criteria for the anatomical material were the surgical interventions on the face, the presence of scars, severe facial deformities or previously implanted esthetic materials. The blood vessels of the heads were stained using a silicone compound according to the method of M. Landofi [16]. Silagerm 7102 silicone compound was used for the preparation of the solution; Silastic LPX 2002 red and Silastic LPX 5019 blue pigments were used for staining, Lasil 81-VF NW (Dow Corning, Wiesbaden, Germany) was used as a hardener, and a benzene solution was used as a solvent. Dissection was performed on each half of the face using standard surgical dissecting instruments, under control with surgical microscope (Leica M220 F12, Leica, Wetzlar, Germany). The skin incision was made along a line running down in front of the ear and parallel to it towards the angle of the mandible. Further, a small incision on the superficial fascia anteriorly to the tragus was extended thereafter up and down, corresponding to the skin incision. After SMAS, blunt dissection was carefully made in the medial direction up to the line between the junction point of the zygomatic arch and the body of the zygomatic bone superiorly and the intersection point of the anterior edge of the masseter muscle and the lower edge of the mandible inferiorly.

In addition, a channel under the superficial fascia of the neck was bluntly dissected downwards and medially. This was necessary to be able to further dissect the SMAS flap towards the neck and move it in the medial direction. On the way to the aforementioned line, four resistance points were identified that corresponded to the most typical location of the ligaments chosen for the study. After that, the SMAS was dissected from the angle of the mandible in the direction toward the midline of the neck. The SMAS flap was flipped in the medial direction. The isolated structures connecting the underlying tissues and the overlying SMAS and their relations were evaluated. Pictures were taken using a Sony Alpha ILCE-6100L (Sony, Tokyo, Japan) camera. The length, origin and insertion, dimensions of the origin site, as well as the structure of the ligaments were assessed. All measurements were carried out during dissection using a standard mechanical curvimeter, ruler, and caliper. The area of the attachment sites of the ligaments was assessed using the Image Pro Plus 6.0 (Media Cybernetics, Rockville, MD, USA) software. The data were entered into individual dissection protocols and processed using “Statistica 6.0” (StatSoft, Tulsa, OK, USA) software. Mean and standard deviation (M ± SD) were counted. 

## 3. Results

In all cases, the zygomatic ligament was identified (100%, *n* = 60). It consisted of three to seven dense connective tissue strands running obliquely in the medial direction. The fibers originated at the lower edge of the origin of the zygomaticus major muscle of the muscle (Figure 1). Some of the fibers were tendon cords originating from the muscular bundles of the zygomaticus major muscle. Similar observations were typical for relatively younger heads (under the age of 71, *n* = 21, 35%). 

In addition, the relationships of the zygomatic ligament with the parotid duct, the transverse facial artery, and the zygomatic branches of the facial nerve were assessed. The parotid duct was running near to the zygomatic ligament in 63.33% (*n* = 38) of observations; at the same time, its fascial sheath was often attached to the lowest cords of the zygomatic ligament. The transverse facial artery was related to the zygomatic ligament only in cases when it accompanied the parotid duct (20%; *n* = 12) or was located above it (38.33%; *n* = 23). The zygomatic branch of the facial nerve, in all observations, was the deepest structure traveling under the zygomatic ligament toward its destination deep in the zygomaticus major muscle. In all cases, the zygomatic ligament cords were running deep into the SMAS and then inserted into the dermis. 

The results of the zygomatic ligament measurement are shown in Table 1. 

The area of the origin of the zygomatic ligament was 1.01 ± 0.26 cm^3^. The mean length of the ligament was 1.15 ± 0.26 cm.

As shown in Table 1, no significant difference in the origin and the length of the zygomatic ligament between the sexes or right and left halves of the face were found (*p* > 0.05).

The upper masseteric ligaments were located more medially, representing two obliquely and horizontally arranged connective tissue laminae extending from the parotideomasseteric fascia. One of the buccal branches of the facial nerve or small collaterals of the transverse facial artery traveled between them (Figure 2). 

At the same time, the parotid duct in cases of its location under the zygomatic ligament (63.33%; *n* = 38) entered the buccal region through the gap between the zygomatic ligament and the upper masseteric ligament. Fully presented upper masseteric ligaments were found in 83.33% (*n* = 100) of observations. In other cases (16.66% *n* = 20), these laminae were thin and could be easily destroyed, including due to blunt dissection. The fibers of this ligament could not be followed up after they were inserted into the fatty tissue of the SMAS layer. It was not possible to estimate the area of the origin due to the small size of the structure. The length of the ligament was significantly smaller than zygomatic ligament (0.68 ± 0.26 cm) (Table 2). 

As follows from the Table 2, no significant difference in the length of the upper masseteric ligament between the sexes or right and left halves of the face were found (*p* > 0.05).

The lower masseteric ligament (100%; *n* = 60) was a connective tissue lamina extending from the parotideomasseteric fascia at the level of the lower third of the anterior edge of the masseter muscle. The part of the connective tissue fibers of this ligament followed from the sheath of the facial vein, in cases of its typical location within this sheath (56.66% *n* = 34) (Figure 3) [17]. The weaving of the lower masseteric ligament as well as the upper one was limited by the superficial fascia, i.e., the SMAS. Similarly to the upper masseteric ligament, in a few cases, the lower masseteric ligament was significantly thinned and easily destroyed by external interventions. The lower masseteric ligament was slightly longer in comparison with the upper masseteric ligament (0.67 ± 0.26 cm), its mean length was 1.12 ± 0.22 cm (Table 3). 

As follows from the Table 3, no significant difference in the length of the lower masseteric ligament between the sexes or right and left halves of the face were found (*p* > 0.05).

The mandibular ligament, unlike the upper and lower masseteric ligament, was clearly identified in all (100%; *n* = 60) observations. The ligament consisted of two rather wide (0.5 ± 0.17 cm) connective tissue laminae originating from the parotideomasseteric fascia at the lower edge of the mandible and from the inner surface of this fascia, along the anterior edge of the masseter muscle. Thus, these two fascial laminae, skirting the sheath of the facial vein and the facial artery obliquely, traveling towards the platysma and the depressor anguli oris muscle, diverging at an angle and merging with these muscles (Figure 4).

This ligament was the longest compared to other ligaments, its mean length is 1.31 ± 0.29 cm.

As follows from the Table 4, no significant difference in the length of the mandibular ligament between the sexes or right and left halves of the face were found (*p* > 0.05).

## 4. Discussion

The results obtained allow us to state the musculo-skeletal-cutaneous nature of the zygomatic ligament, which, unlike the other three ligaments, is a true ligament, which can originate from the zygomaticus major muscle. This feature of the ligament suggests re-evaluating its age-related changes, since age-related degeneration of the zygomaticus major muscle, in addition to bone resorption and stretching of the ligamentous apparatus of the face, contributes to the aging mechanism. On the other hand, this gives the opportunity to affect the structure of the zygomatic ligament through the influence on the zygomaticus major muscle. In addition, it does not appear relevant to mention any relations between the zygomatic ligament and the parotid duct located below it. Despite the close location of these two anatomical structures, they lie at different depths and, moreover, the dependence of the position of the parotid duct on the position of the zygomatic ligament would contradict the detected variants of the duct’s course [18]. Thus, the possible location of the parotid duct in the space between the zygomatic and upper masseteric ligaments remains one of the options for its course. As for the upper masseteric ligament, the presented images (Figure 2 and Figure 3) accurately show the nature and structure of this ligament, which coincides with the following description of the zygomatic ligament by McGregor: “As soon as you cut through this patch, you lose the protection of the parotid fascia on your deep side and you will see loose fat with branches of the facial nerve looking at you, hopefully intact…running through the loose fat, a little deeper and a little caudad, is the parotid duct…as stated above, I have for years emphasized that this patch is a warning sign asking the surgeon to be aware of the road ahead” [12]. In this regard, it can be disputed that the so-called McGregor’s patch is not the zygomatic but the upper masseteric ligament. The upper and lower masseteric, and the mandibular ligaments, in our opinion, should be combined into a single group of fascio-SMAS ligaments connecting the anteromedial edge of the parotideomasseteric fascia with the anteromedial margin of the SMAS and its superficial fascia. This point of view, in our opinion, allows us to take a different look at the relations between SMAS and the parotideomasseteric fascia, since the parotid fascia in this case, with its anterior edge and the described ligamentous apparatus, acts as additional points of fixation of the superficial fascia and platysma muscle. Thus, the parotideomasseteric region turns out to be strongly isolated from the buccal and zygomatic regions, and fascio-SMAS ligaments can perform their barrier functions. In addition, the false nature of these ligaments, since none of them are directly inserted into the skin, indicates their lesser participation in the described mechanisms of facial aging [1], giving greater importance to bone resorption and SMAS degradation.

## 5. Conclusions

Despite the common location of the zygomatic, upper and lower masseteric, as well as the mandibular ligaments along the anteromedial border of the parotideomasseteric region, it is necessary to clearly distinguish these ligaments, as they belong to the two different groups. The first group includes a true, musculo-skeletal-cutaneous zygomatic ligament that plays an important role in fixing the skin of the face; another group of false fascio-SMAS ligaments fixes the superficial fascia and platysma muscle. The difference in the attachments and structure of these ligaments should certainly explain the diverse role of muscle degeneration and stretching of the ligaments for the mechanisms of face aging due to the gravitational ptosis of facial soft tissues.

## Figures and Tables

**Figure 1 medicina-60-00681-f001:**
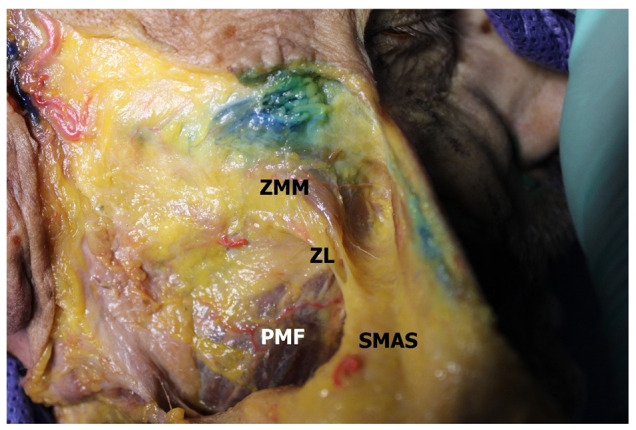
Retaining ligaments of the face. ZMM—zygomatic major muscle; ZL—zygomatic ligament; PMF—parotideomasseteric fascia; SMAS—superficial musculoaponeurotic system.

**Figure 2 medicina-60-00681-f002:**
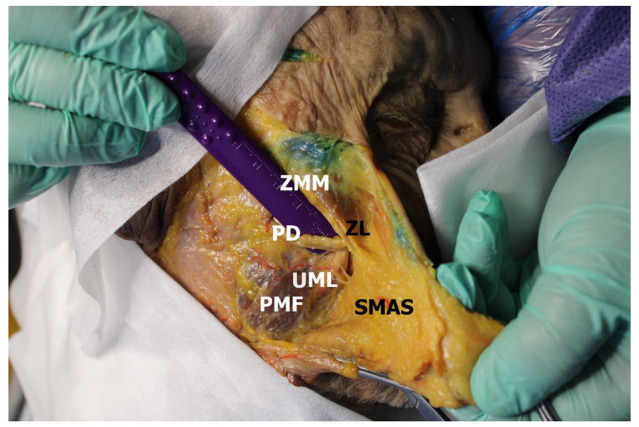
Retaining ligaments of the face. ZMM—zygomatic major muscle; ZL—zygomatic ligament; PMF—parotideomasseteric fascia; SMAS—superficial musculoaponeurotic system; PD—parotid duct; UML—upper masseteric ligaments.

**Figure 3 medicina-60-00681-f003:**
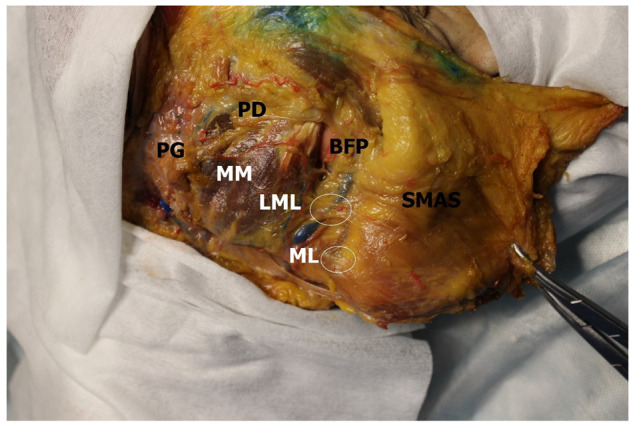
Retaining ligaments of the face. PG—parotid gland; PD—parotid duct; BFP—buccal fat pad; MM –masseteric muscle; SMAS—superficial musculoaponeurotic system; LML—lower masseteric ligaments; ML—mandibular ligament.

**Figure 4 medicina-60-00681-f004:**
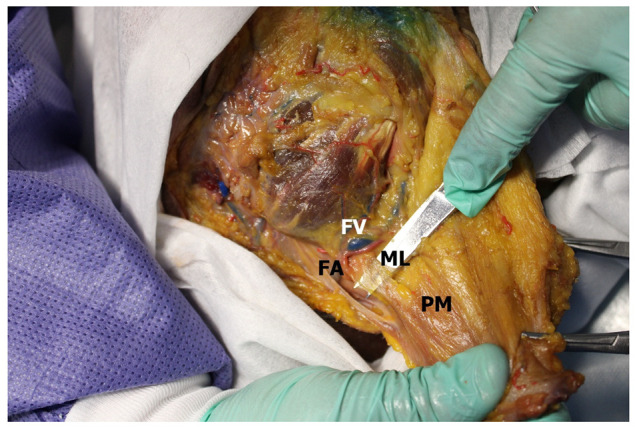
Retaining ligaments of the face. FV—facial vein; FA—facial artery; ML—mandibular ligament; PM—platysma muscle.

**Table 1 medicina-60-00681-t001:** The area of origin site and the length of the zygomatic ligament (M ± SD).

Parameters	Males		Females		Total (%)	
	Right (*n* = 15)	Left (*n* = 15)	Right (*n* = 15)	Left (*n* = 15)	Right (*n* = 30)	Left (*n* = 30)
area (cm^3^)	1.00 ± 0.26	1.04 ± 0.27	0.98 ± 0.28	1.01 ± 0.25	0.99 ± 0.27	1.02 ± 0.26
length (cm)	1.16 ± 0.28	1.18 ± 0.24	1.17 ± 0.24	1.09 ± 0.27	1.17 ± 0.26	1.14 ± 0.25

**Table 2 medicina-60-00681-t002:** The length of the upper masseteric ligament (M ± SD).

Parameters	Males		Females		Total (%)	
	Right (*n* = 27)	Left (*n* = 27)	Right (*n* = 23)	Left (*n* = 23)	Right (*n* = 50)	Left (*n* = 50)
length (cm)	0.66 ± 0.2	0.69 ± 0.25	0.70 ± 0.28	0.67 ± 0.26	0.67 ± 0.27	0.68 ± 0.25

**Table 3 medicina-60-00681-t003:** The length of the lower masseteric ligament (M ± SD).

Parameters	Males		Females		Total (%)	
	Right (*n* = 15)	Left (*n* = 15)	Right (*n* = 15)	Left (*n* = 15)	Left (*n* = 15)	Right (*n* = 15)
length (cm)	1.08 ± 0.21	1.15 ± 0.22	1.16 ± 0.24	1.07 ± 0.23	1.12 ± 0.23	1.11 ± 0.22

**Table 4 medicina-60-00681-t004:** The length of the mandibular ligament (M ± SD).

Parameters	Males		Females		Total (%)	
	Right (*n* = 15)	Left (*n* = 15)	Right (*n* = 15)	Left (*n* = 15)	Right (*n* = 15)	Left (*n* = 15)
length (cm)	1.32 ± 0.31	1.31 ± 0.27	1.36 ± 0.27	1.28 ± 0.31	1.34 ± 0.29	1.29 ± 0.29

## Data Availability

Data are contained within the article.

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
