# Peer review of "Clinical Anatomy of the Ligaments of the Face and Their Fundamental Distinguishing Features"

_medicina, 2024, doi:10.3390/medicina60050681_

Round 1

Reviewer 1 Report

Comments and Suggestions for Authors

Dear Authors, 

You made a great work! I suggest some improvements before acceptance.

The paper is an anatomical ex vivo study on the clinical anatomy of the ligaments of the face and their fundamental distinguishing features.

The Authors made a great work in terms of methodology and the paper sounds scientific and well written.

The abstract is well written, complete and summary in its various aspects. The keywords are complete and appropriate.

It is always of great interest to read more articles regarding cadaver anatomy, precisely because of this fantastic opportunity for growth and development of knowledge in light of new technologies and capabilities. From this point of view the introduction is absolutely well written and linear.

The materials and methods section is simple and schematic, providing the necessary information for this type of manuscript.

The results section is well described, I suggest the authors work on the layout of the tables and the section in general, the information reported in this section is sufficient.

The discussion is well written, complete, and I believe that the Authors have adequately developed the problem, working in an absolutely orderly and well-organized manner.

Conclusions are concise and clear.

Bibliography should be formatted respecting the journal’s requirements and no improper citations are evidenced.

Figures and labels are clear and easy to comprehend.

English is clear and easy to understand, I suggest some minor revisions. 

Comments on the Quality of English Language

I believe that the more discursive sections of the manuscript can be written more correctly.

Author Response

Hello, dear reviewer!

We thank you for your work with our scientific work and for the comments you left.

We revised the list of references, link 15 was fixed and added doi to all available references

Reviewer 2 Report

Comments and Suggestions for Authors

In this paper, we observed the zygomatic ligament, upper masseter ligament, lower masseter ligament, and mandibular ligament. We measured the length etc. and considered the relationship with SMAS.

Please check the following points.

In L.17, the above ligaments are described abstractly without any prior explanation. I think it would be good if you could explain a little bit about ligament retention.

L.90 says that blunt dissection was performed carefully, but was a surgical microscope used? Please provide details if used.

L.103 states that the length of the origin, the origin and insertion, and the dimensions were measured, but only the origin of the zygomatic ligament is included in the results, and in other cases only the length is measured. Included in the results. result.

L.109 says that you used the student-t test, but the results compare the relationship between the data of the zygomatic ligament, upper masseter ligament, lower masseter ligament, and mandibular ligament, so it is better to use that. Probably. Analysis of variance. Isn't that a good thing?

Regarding the statement that the zygomatic ligament in L.113 is derived from the zygomaticus major muscle, I think it would be better to state the frequency of occurrence as "in all cases."

L.134 says "area of the zygomatic ligament origin", but I think it would be better to describe the details of the measurement method in the materials and methods.

It says that L.166 is a little longer, but it's just that the average value is a little larger. It is better to include the results of statistical tests.

For the two fairly wide (0.5 ± 0.17 cm) connective tissue layers in L.176, no specific measurement method (site, tool) was described.

In L192, the zygomatic ligament is considered to be the true ligament, but if there is evidence that it originates from the zygomaticus major muscle, I think it would be better to clearly state in the results that it originates from the zygomaticus major muscle.

L.229 states that they are a separate group, but I think the basis for this is unclear as this paper does not describe the relationship between facial muscles and ligaments other than the relationship between the platysma muscle and mandibular ligament.

Author Response

Dear reviewer!
Thank you for your attention to our work and the valuable comments you gave us.

Below I would like to answer your questions point by point and as they are presented in the text
1. n L.17, the above ligaments are described abstractly without any prior explanation. I think it would be good if you could explain a little bit about ligament retention.

Answer: added names of connections on lines 17-18

2. L.90 says that blunt dissection was performed carefully, but was a surgical microscope used? Please provide details if used.

Answer: added information about the used microscope and tools 89-90

3. L.103 states that the length of the origin, the origin and insertion, and the dimensions were measured, but only the origin of the zygomatic ligament is included in the results, and in other cases only the length is measured. Included in the results. result.

Answer: it is indicated what was done during dissection

4. L.109 says that you used the student-t test, but the results compare the relationship between the data of the zygomatic ligament, upper masseter ligament, lower masseter ligament, and mandibular ligament, so it is better to use that. Probably. Analysis of variance. Isn't that a good thing?

Answer: the phrase about the Student's test remained from the draft version of the article, we have now removed it, because data was calculated in the Statistic software package

5. Regarding the statement that the zygomatic ligament in L.113 is derived from the zygomaticus major muscle, I think it would be better to state the frequency of occurrence as "in all cases."

Answer: clarification added (100%, n=60).

6. L.134 says "area of the zygomatic ligament origin", but I think it would be better to describe the details of the measurement method in the materials and methods.

Answer: we have added information in the "materials and methods" section about measurements during dissection

7. It says that L.166 is a little longer, but it's just that the average value is a little larger. It is better to include the results of statistical tests.

Answer: Added clarifying parameters

8. The lower masseteric ligament was slightly longer in comparison with the upper masseteric ligament (0.67±0.26 cm), its mean length was 1.12±0.22 cm (Table 3).
For the two fairly wide (0.5 ± 0.17 cm) connective tissue layers in L.176, no specific measurement method (site, tool) was described.

Answer: we have added information to the materials and methods about measurements during dissection, because We did not measure each structure specifically and separately. A standard method was used, which allows for a comprehensive assessment of tissues without strong errors

9. In L192, the zygomatic ligament is considered to be the true ligament, but if there is evidence that it originates from the zygomaticus major muscle, I think it would be better to clearly state in the results that it originates from the zygomaticus major muscle .

Answer: 198-199 wrote that which can originates from the zygomaticus major muscle.

10. L.229 states that they are a separate group, but I think the basis for this is unclear as this paper does not describe the relationship between facial muscles and ligaments other than the relationship between the platysma muscle and mandibular ligament.

Answer: It is known that the mandibular ligament is located in the lateral region of the face near the masticatory muscle, so we do not separate it into the structural complex of the supporting ligaments of the facial muscles. The purpose of our study was specifically the ligaments, and not the facial muscles, so such conclusions cannot be drawn. We thank the reviewer for this issue, which we will definitely explore in the next study.